# Laser Welding Penetration Monitoring Based on Time-Frequency Characterization of Acoustic Emission and CNN-LSTM Hybrid Network

**DOI:** 10.3390/ma16041614

**Published:** 2023-02-15

**Authors:** Zhongyi Luo, Di Wu, Peilei Zhang, Xin Ye, Haichuan Shi, Xiaoyu Cai, Yingtao Tian

**Affiliations:** 1School of Materials Science and Engineering, Shanghai University of Engineering Science, Shanghai 201620, China; 2Shanghai Collaborative Innovation Center of Laser of Manufacturing Technology, Shanghai 201620, China; 3School of Materials Science and Engineering, Shanghai Jiao Tong University, Shanghai 200240, China; 4State Key Laboratory of Advanced Welding and Joining, Harbin Institute of Technology, Harbin 150001, China; 5Department of Engineering, Lancaster University, Lancaster LA1 4YW, UK

**Keywords:** pulsed laser welding, acoustic emission, keyhole dynamics

## Abstract

In-process penetration monitoring of the pulsed laser welding process remains a great challenge for achieving uniform and reproducible products due to the highly complex nature of the keyhole dynamics within the intense laser-metal interactions. The main purpose of this study is to investigate the feasibility of acoustic emission (AE) measurement for penetration monitoring based on acoustic wave characteristics and deep learning. Firstly, a series of laser welding experiments on aluminum alloys were conducted using high-speed photography and AE techniques. This allowed us to in-situ visualize the complete keyhole dynamics and elucidate the generation mechanism of acoustic waves originating from pressure fluctuations at the keyhole wall. Then, an adaptive time-frequency technique namely VMD (Variational Mode Decomposition) was proposed to characterize the acoustic energy distribution among the nine subsignals with low-frequency and high-frequency components under different welding penetrations. Lastly, a novel hybrid model combing CNN (Convolutional Neural Network) and LSTM (Long Short Term Memory) was designed to deeply mine the spatial and temporal acoustic features from the extracted frequency components. Extensive experiments demonstrate that our proposed approach yields a remarkable classification performance with a test accuracy of 99.8% and a standard deviation of 0.21, which obtains a high recognition rate. This work is a new paradigm in the digitization and intelligence of the laser welding process and contributes to an alternative way of developing an efficient end-to-end penetration monitoring system.

## 1. Introduction

Aluminium alloys have the advantages of high specific strength, good corrosion resistance, and high thermal conductivity and are widely used in aerospace and automotive manufacturing [1]. 6XXX series aluminium alloys are widely used in the automotive manufacturing industry, usually in the fields of battery manufacturing and automotive chassis manufacturing by joining in a welding way.

However, due to the poor heat resistance and strong thermal conductivity of aluminium alloys, defects are very easy to form in welding [2]. Traditional arc welding for welding aluminium alloys has large welding deformation and low productivity, while electron beam welding of aluminium alloys requires a guaranteed vacuum environment, resulting in high costs and low efficiency. However, laser welding has a high energy density and high welding speed, which allow more precise control of the heat input. Therefore, laser welding in aluminium alloys has a high energy density, low heat input, low welding distortion, and a large depth to width ratio.

Therefore, laser welding processes with such advantages as high energy, high precision, and high efficiency have been widely applied in high-end industry fields such as aerospace, automotive, shipbuilding, etc. [1]. With its precise regulation of laser energy, the pulsed mode laser welding technique is particularly suitable for joining thin aluminium alloys with high reflectivity and high thermal conductivity [2]. In general, the combination of pulse frequency, pulse width, peak power, and other parameters can directly affect the welding process stability. Particularly, the intense interaction between the high-energy laser beam and aluminium alloy easily drills an extremely unstable vapor capillary (referred to as a “keyhole”) into the workpiece. As such, any tiny change of operation conditions can easily produce the instability phenomena of melt pool morphology and intense keyhole oscillation, which may easily cause the welding defects occurring in the welding process [3]. Therein, insufficient penetration weakens the mechanical properties of the welded parts, while excessive penetration can be characterized by sagging, holes, and bum-through. 

To understand the keyhole dynamics and further improve the product quality, a series of in-situ monitoring approaches, including thermal [4,5], photoelectric [6], and optical sensors [7,8], have been proposed in industry and academia. The design idea of mainstream monitoring solutions lies in the utilization of the phenomenon of keyhole formation, which carries various types of valuable information about weld quality. It is worth noting that acoustic waves play a key role in driving pores near the keyhole tip far enough away from the large thermal gradient field around the keyhole during the laser melting process [9]. 

Therefore, a promising acoustic emission (AE) monitoring system has received much attention recently due to its cost-efficient, in-line, and non-invasive measurement [10]. It can capture the strong acoustic wave generated from the laser-material interactions, propagating inside the workpiece and atmosphere. Due to this nature, on-line AE measurement can detect the pressure fluctuation generated by the molten pool pulsation, plasma/metal vapor generation, thermal stress, and keyhole oscillation, which are closely related to the depth of weld penetration. However, the key challenge in the development of a reliable and cost-effective acoustic monitoring system is the intricate dependency between the process parameters, the acoustic signatures, and the weld quality. A detailed investigation of the generation mechanism of acoustic waves and its relation to welding quality has not yet been assessed conclusively. 

(1)Acoustic emission monitoring

To overcome this issue, Gu et al. [11,12] applied a frequency analysis and a statistical approach to deal with the airborne sound signals acquired in the CO_2_ laser welding process. The results suggest that the band of frequency components is well associated with the keyhole oscillation, and these frequencies are distinctive when the welds are fully penetrated. Huang et al. [13,14] used a spectral subtraction method to reduce the background noise in the acquired acoustic signals and then applied a power spectrum method (Welch-Bartlett) to analyze the frequency characteristics of the acoustic signals. The relationship between acoustic features and various penetration depths can also be established based on time- and frequency-domain processing. Recently, Yusof et al. [15,16] extracted the time-frequency characteristics of the collected AE signals based on synchrosqueezed wavelet analysis during pulsed laser welding experiments and statistically correlated them with two process regimes, including half-penetration and full-penetration status. 

To sum up, the observation of distinct AE signals could provide a base for developing a diagnostic/analytical technique for laser welding processes, and it also underpins the existing high potential of AE monitoring in laser processes. Although previous studies have shown that acoustic signals can effectively monitor the laser welding of mild steel, stainless steel, and high-strength steels, there is limited literature on the study of acoustic signals during the pulsed laser welding of aluminum alloys. Moreover, pulse mode laser welding could possibly emit non-stationary period acoustic signals when the pulsed laser beam intermittently interacts with the workpiece over very short time intervals. How to accurately extract the useful information for reflecting the keyhole oscillation and welding stability independently remains a challenge and warrants further research. Consequently, a good prior knowledge of the laser welding process is required to explain the acoustic generation mechanism and acoustic emission trends. 

(2)Machine learning modeling

Based on the afore-cited research, it could be summarized that much of the research up to now has attempted to reveal the significant correlation between the weld quality and the acoustic signatures. Basically, an accurate and quantitative classification of the weld penetration is also imperative as it could directly give feedback to the control system in the automated welding process [17,18]. Wu et al. [19] pointed out that the key term “monitoring” is applied to indicate in-situ information collection, feature extraction, and process modeling. Multiple approaches have been proposed, which are mostly based on analytic/numerical modeling via AE measurements and process data. Due to the high complexity and dynamic nature of laser-metal interactions, a large number of factors affect the final result of repetitive pulsed laser irradiation, and a wide range of relevant physical processes complicate a detailed analysis. Moreover, establishing a complex theoretical or numerical model is rather time-consuming and not suitable for real-time monitoring and quality control.

Therefore, proper acoustic in-process monitoring associated with data-driven machine learning (ML) and feature extraction is needed to increase the productivity and repeatability of laser welding. To our knowledge, a common ML-based algorithm, namely artificial neural networks (ANNs), is capable of dealing with complex databases, and has been widely used to perform tasks such as nonlinear approximation [20], prediction, and classification [21] in a dynamic and complicated industrial process, especially in the field of laser welding. For instance, Huang et al. [14] proposed a neural network and multiple regression method to characterize the depth of weld penetration based on the acquired signatures. The results show that the acoustic signatures can predict the penetration depth well under different laser welding parameters. Lee et al. [22] presented an in-process monitoring scheme for pulsed Nd:YAG laser spot welding based on the AE technique. The amplitudes of three different frequency ranges (100–200, 200–300, and 300–500 kHz) are used as input vectors, and three different process conditions (unsuccessful, successful, and over-welding) are clearly distinguished by utilizing the ANN-based classification model. However, the shallow ANNs usually require feature extraction of acoustic signals in time or frequency domain, which could not completely reflect the dynamic changes of the melt pool and the keyhole under non-stationary and complex conditions of the AE events. 

To overcome the above difficulty, a recently developed convolutional neural network (CNN) is emerging as a promising deep learning (DL) framework since it can automatically learn and characterize the non-linear AE signatures through the multi-layer stacks. For instance, Shevchik et al. [23] studied the feasibility of using acoustic emission (AE) and Spectral Convolution Neural Network (SCNN) for in-situ and real-time quality monitoring by using the energy in narrow-frequency acoustic features. Pandiyan et al. [24] extracted the AE spectrograms using wavelet transforms and applied two CNN architectures, including VGG-16 and ResNets-18, to predict the build quality, including balling, LoF pores, conduction mode, and keyhole pores, during the laser additive manufacturing. 

However, the current CNN methods are limited in exploring the spatial features of the two-dimensional AE signal spectrogram. There are few studies investigating the temporal characteristics of the one-dimensional (1-D) AE signal during the dynamic laser welding process. The newly developed DL framework, called long-short term memory (LSTM), has several advantages in dealing with time information in time-series AE signals. Since the CNN can extract much greater high-level spatial features, recent research attempted to combine the CNN and LSTM models to in-depth mine the spatio-temporal characteristics of signal data, in order to enhance the predicting accuracy and reliability. 

For example, Zhang et al. [25] proposed a dynamic model that combines CNN and LSTM networks to account for the detected AE signals, which can accurately monitor the surface quality during laser shock peening. Shi et al. [26] developed a novel deep learning algorithm (BiConvLSTM) that combines CNN and LSTM networks for planetary gearbox fault diagnosis under different operating conditions. The BiConvLSTM can automatically and simultaneously extract spatio-temporal features from the condition monitoring data collected by three accelerometers and one tachometer sensor. Song et al. [27] developed a CNN-LSTM bearing fault diagnosis model using the VMD (Variational Mode Decomposition) technique to reconstruct the bearing signals, which was then fed into a CNN-LSTM network to extract spatio-temporal features simultaneously and obtain a good diagnostic result. However, there are no investigations in the state of the art yet that correlate the acoustic characteristics to keyhole dynamics and weld penetration depth based on CNN-LSTM. Driven by the intrinsic relationship between acoustic characteristics and welding quality, our study develops a hybrid CNN-LSTM model for penetration prediction to provide a new approach for in-situ monitoring of the laser welding process. The main contributions to this work are as follows: Through investigating the correlations between the transient nature of the subsurface keyhole and the acoustic signals, the AE technique could better characterize the keyhole dynamics and weld penetration. It will also provide important guidance for understanding the complicated interaction mechanisms between the pulsed laser radiation and aluminum alloy and can be extended to other laser-based manufacturing scenarios.In contrast to traditional time- or frequency-domain processing for non-stationary AE signals, an adaptive time-frequency technique called VMD was proposed, which can accurately distinguish between low-frequency and high-frequency components for better describing the weld penetration.A novel CNN-LSTM hybrid model was proposed to deeply mine the spatial and temporal acoustic features from the extracted frequency components. It can improve the penetration predicting performance of AE sensing and provide a potential and reliable monitoring technology for the dynamic laser welding process.

The framework of this paper is organized as follows: Section 2 introduces the experimental setup and process analysis; Section 3 illustrates the AE frequency component extraction process using the VMD; Section 4 presents the spatio-temporal CNN-LSTM model for predicting the weld quality; Section 5 illustrates the performance results compared with other algorithms; the main conclusions and future study are stated in Section 6. 

## 2. Methodical Approach

### 2.1. Experimental Setup and Data Acquisition

To reveal the relationship between AE signals and keyhole characteristics as well as the weld penetration during pulsed laser welding, a complete experimental platform was created, including: (i) a multi-information acquisition system, (ii) a pulsed mode laser welding system, and (iii) a traveling mechanism system, as illustrated in Figure 1. In our work, the laser welding system mainly contains a rectangular pulsed fiber laser device (TRUPULSE 556, *λ* = 1064 nm) with a 10 kW maximum output power. A laser head (Trumpf BEO D70, Ditzingen, Germany) with an inclination angle of 5° was applied to avoid the back reflection damaging the delivery optics, particularly for welding highly reflective aluminum alloys. Pure argon was used as the shielding gas, and a rectangular pulse type was selected with a beam diameter of 0.6 mm at the focal position. The traveling mechanism system was applied to drive the workpiece while the laser head remained fixed in the vertical position. 

To gain a better understanding of the AE technique and to broaden its application for estimating different weld penetration, it is necessary to test its capability with a wide range of laser parameters. Thus, a series of different bead-on-plate experiments were conducted on aluminum alloy (6061) plates by modifying the laser power. Other parameters remained unchanged, as listed in Table 1. This procedure was repeated in order to create a data basis for the evaluation of the correlation between achieved penetration depth and AE with respect to the adjustable welding parameters. After laser welding, the weld’s quality was characterized postmortem by means of metallographic testing using an optical microscope setup.

The multi-information synchronous acquisition system consists of two subsystems: visual sensor and the ABAE sensor (BSWA TECH, Beijing, China, MA231 pPreamplifier). Therein, the visual sensing system contains a high-speed camera (PHANTOM VEO 710L) with a band-pass filter (810 nm) and an auxiliary illuminant (CAVILUX HF, 810 nm ± 10 nm), which could efficiently reduce the existing interferences of spatters and plasma plumes. The sampling frequency was set to 10,000 fps, and the exposure duration was 1 μs. The original lateral keyhole image was 512 × 384 pixels with a resolution of 0.022 mm/pixel. In addition, a butt-joint configuration consisting of an aluminum alloy (6061) with size of 100 mm (length) × 50 mm (width) × 2 mm (height), and transparent heat-resistant quartz glass with size of 100 mm (length) × 5 mm (width) × 2 mm (height), was constructed to directly observe the keyhole dynamics inside the melt pool through the high-speed imaging during the laser welding process. The ABAE signals (airborne sound pressure waves) were collected using a condenser microphone (BSWA TECH, MA231 Preamplifier) with a sampling frequency of 20 kHz. The microphone was oriented at an angle of 45° with the horizontal axis, and the distance between the laser-metal interaction and the microphone was 50 mm. The national standards for the chemical composition [28] and mechanical properties [29] of 6061 aluminium alloys are given in Table 2.

### 2.2. Detected AE Signals and Process Analysis

It is demonstrated that the weld penetration is of importance to weld formation/joint quality, and the backside weld width is usually proposed to evaluate the penetration status quantitatively. According to the actual weld widths, the degree of weld penetration can be categorized into three types, including: (i) none-penetration (NP), (ii) partial-penetration (PP), and (iii) full-penetration (FP), as depicted in Table 1. Figure 2 displays the transverse cross sections (perpendicular to the laser path) of the fusion zone and the corresponding AE signals detected under different penetration statuses. We can find that the AE signals mainly change in the time-domain in terms of amplitude fluctuation. It also consists of a lot of impulses, referring to the pulsed AE events. The cycle of each pulsed AE event is 0.10 s because the laser pulse frequency applied in our experiment is 10 Hz. When the amplitude distribution of a pulsed AE event in the time-domain is uniform, the corresponding welding process is relatively stable, which creates a better appearance of cross-section morphology. When the laser heat input is always less than the energy threshold for forming a penetration hole, the intensity of the acoustic signal intensity remains small (less than 2 V) under the non-penetration status. 

As the higher laser energy is gradually delivered to the lower surface of the workpiece, the full-penetration hole may not necessarily be formed in each pulse period, which refers to critical-penetration status. Some pulse periods of acoustic signal intensity increased to exceed 2 V, and other pulse periods of signal intensity were less than 2 V. As the laser heat input further increased to significantly exceed the energy threshold, the full-penetration status easily occurred and the acoustic signal intensity reached a maximum value (larger than 3 V) under a stronger laser-beam illumination.

In general, the pulsed AE events can effectively reflect the keyhole dynamics of the pulsed laser affecting the workpiece during the laser welding process. The process-based definitions of various penetration modes come directly from time-resolved imaging of keyhole morphology. Taking a full-penetration weld as an example, Figure 3 displays the dynamic evolution of the keyhole and a single pulse AE event extracted from the complete AE signal during pulsed laser mode. Under the high instantaneous energy of the laser beam in Phase 1, a concave quickly appeared within a very short time due to a strong recoil momentum from the vaporization of the liquid surface. The intensity of the AE signal also began to increase. With an increase in laser energy in Phase 2, the laser fluence is sufficient to vaporize the metal, generating a large recoil pressure that opens a deep and slender vapor depression (referring to a gas-vapor keyhole) based on multiple reflections, which substantially enhances the laser energy absorption by the surrounding melt pool. Meanwhile, the amplitude of the AE signal increased significantly to a maximum value, which indicates that the formation process of the keyhole releases intense AE energy. From Phase 3 to 4, the laser absorption and reflection occurred due to its interaction with the vaporized metal; only part of the laser energy was absorbed by the molten metal. Thus, the depth of the keyhole decayed with time, and the amplitude of the AE signal began to decrease accordingly. After the pulse laser termination, the keyhole disappeared quickly over a very short interval of time and reappeared until the next laser pulse started. Meanwhile, the intensity of the AE signal also degraded rapidly due to the absence of laser energy, as illustrated in Phase 5. 

By discussing the correlations between the transient nature of the subsurface keyhole and the AE signal, it is indicated that the acoustic emission could provide a wealth of information about the keyhole dynamics under the complicated interaction between the repetitively pulsed laser radiation and the aluminum alloy workpiece. To fully implement the acoustic monitoring system and manufacture defect-free welded products, we need a more comprehensive understanding of the acoustic source and its internal relationship to the weld quality.

### 2.3. Generation Mechanism of Acoustic Wave

As depicted in Figure 4a, a high-energy Gaussian-profiled laser beam (*I* ≥ 1 MW/cm^2^) focusing on the surface of a workpiece creates a thin and narrow capillary called a keyhole in a moving material. According to the hydrodynamic theory [30], the dynamic pressure balance at the keyhole wall is a premise of the stable keyhole that exists in the deep-penetration laser welding process. Significantly, the slight variation of the keyhole shape and size (depth *d* and exit radius *R*) could easily cause the pressure to fluctuate at the keyhole wall. 

To keep the keyhole open and stable, the various driving pressures at the keyhole wall should satisfy a balance equation of pressure Δp=0, as follows [30]:(1)$$\Delta p=p_{\gamma }(r)+p_{abl}(T)+p_{l}+p_{g}$$
(2)$$\mathrm{Surface}\;\mathrm{tension}\;\mathrm{pressure}:\;p_{\gamma }=\frac{\gamma }{R}$$
(3)$$\mathrm{Ablation}\;\mathrm{pressure}:\;P_{abl}(T_{s})=\frac{I_{r}}{L_{v}}(\frac{\kappa \cdot R\cdot T_{s}}{M})$$
(4)$$\mathrm{Hydrodynamic}\;\mathrm{pressure}:\;p_{l}=\frac{1}{2}\cdot \rho \cdot v^{2}$$
(5)$$\mathrm{Hydrostatic}\;\mathrm{pressure}:\;p_{g}=\rho \cdot g\cdot d$$
where *P_γ_* denotes the surface tension pressure induced by the molten metal around the keyhole, *γ* the surface tension coefficient, *P_abl_* the ablation pressure induced by the material evaporation, *L_v_* the latent heat of vaporization, *I_r_* the laser intensity for material vaporization, κ the thermal diffusivity, *T_s_* the keyhole surface temperature, *M* the molar mass of aluminium alloy, *ρ* the fluid density and *g* the gravity acceleration, *v* the flow velocity of the liquid metal fluid, and *d* the depth of the keyhole. 

It is demonstrated that the equilibrium values of the keyhole sizes depend on the Gaussian laser energy distribution on the keyhole wall and material properties. With the increase in temperature at the bottom of the vapor depression, the laser absorption also increases abruptly, disrupting the force balance at the local vapor-liquid interface. The keyhole then becomes unstable and grows rapidly in depth. Due to the instability of the keyhole itself, any small deviation of the keyhole depth and radius from their equilibrium values leads to a pressure fluctuation and keyhole oscillation. Significantly, the response of keyholes to pressure fluctuations has been predicted to involve the excitation of a number of normal modes at characteristic frequencies [31]. The eigenfrequency *f*_0_ of keyhole oscillation can be calculated by Equation (6):(6)$$f_{0}^{2}=\frac{\gamma d^{2}+2\rho gRd^{3}-3\gamma R^{2}}{\ln3\times \delta (3R^{5}+d^{2}R^{2})}$$

It can be seen that the eigenfrequency *f*_0_ is closely related to the keyhole depth *d* and radius *R* under the dynamic balance of the keyhole. Meanwhile, the keyhole oscillations (e.g., radial, axial, and azimuthal) induced by the interaction between the different laser energy intensities and materials could generate shock acoustic waves in different ranges of amplitude/frequency, which could be reflected in the acoustic intensity and spectrum. In addition, the acoustic emission in the audible frequency range is also likely to be related to the vapor plume ejection from the keyhole outlet. The keyhole-induced vapor plume compresses the ambient atmosphere and generates a high air pressure. It will further produce an acoustic wave with a large overall amplitude, depending on the plume size, which will rely on the laser energy input. Particularly, the expanding plume dynamics are influenced by the keyhole fluctuations caused by distortion of the pressure balance. The vapor ejection and the plasma size are modulated by the keyhole oscillation under the dynamic pressure fluctuations. Thus, the oscillation of the keyhole itself is considered a potential point acoustic source, and the molten pool acts like a speaker diaphragm, generating acoustic waves in the surrounding atmosphere.

To verify the rationality of the acoustic source mechanism, the keyhole depth was extracted under different laser powers with a common imaging-processing algorithm according to our previous studies [32,33]. From a series of variation curves of keyhole depths in each laser pulse (see Figure 4b), we found that: (1) when the laser energy is lower (3 kW), the depth variation of keyhole increases from 0.7 mm to 1.5 mm. The average depth (see the solid red line) also changes gradually, which is close to 1.2 mm. Consequently, the acoustic signal intensity is lower under the none-penetration status, as depicted in Figure 2a. This is because the oscillation behavior of the keyhole is not intense under the relatively smooth changing pressure acting on the keyhole wall; (2) when the laser energy is further increased, the keyhole depth dynamically changes from 1.8 mm to 3.0 mm, and the acoustic signal intensity also increases significantly under the full-penetration status (see Figure 2c). It is indicated that the evolution process of a penetrated keyhole releases intense AE energy due to a more intense keyhole oscillation behavior. Based on the theoretical and experimental analysis, the AE characteristics are a strong function of various influencing factors, including the material properties, experimental conditions (laser power, pulse duration, and frequency), and acoustic waves induced by the keyhole oscillation. 

For better correlating the acoustic wave characteristics with the various weld penetrations, revealing the amplitude and spectrum distribution of acoustic signals with time-frequency analysis is indeed crucial to enhance the potential adoption of the AE technique, which will be discussed in the next section. 

### 2.4. Time-Frequency Analysis of AE Signals

Since the spectrum property of the AE signal is closely related to the keyhole oscillation and vapor plume expansion generated by the interaction between the pulsed laser beam and materials, the frequency-domain analysis is a more accurate and efficient analytical approach, providing additional interesting insights compared to the time-domain. Additionally, the frequency-domain features on the specific frequency band are not sensitive to the environment and the machine noises from other frequency bands. In our study, a Fast Fourier Transform (FFT) algorithm was applied to extract the frequency characteristics of the AE signal segments, as shown in Equation (7):(7)$$X(k)=\sum_{n=0}^{N-1}x(n)(\cos2\pi k\frac{n}{N}-j\sin2\pi k\frac{n}{N})\;(k=0,1,2,\ldots ,N-1)$$
where *X*(*k*) is the calculated data after FFT processing, *x*(*n*) is the recorded acoustic signal, and *N* is the sample points (*N* = 20 k). To directly reflect the distribution of frequency-domain characteristics over time, the acoustic signals belonging to different penetration states were transformed into a 2-D spectrogram using the short-time Fourier transform (STFT). The corresponding “time-frequency-amplitude” information is investigated in combination with the keyhole behavior. By shifting the window function *h*(*t*) on the time axis, the acoustic signal was analyzed segment by segment to obtain a local spectrum of the acoustic signals. The STFT of the raw signal *x*(*t*) is defined as:(8)$$\begin{array}{c} STFT\left(t,f\right)=\int_{-\infty }^{+\infty }x\left(\tau \right)h\left(\tau -t\right)e^{-j2\pi f\tau }d\tau \end{array}$$

Figure 5 depicts the formation and dynamic evolution of the keyhole and the corresponding depth variation, as well as the 2-D spectrograms under different weld penetrations. It is found that the time-frequency spectrograms transformed by the STFT algorithm can visually display an obvious pulse component describing the pulse laser action over time. Moreover, the spectrum intensity of a low-frequency band (less than 2 kHz) under various weld penetration is almost equal, while the spectrum intensity of a high-frequency band under PP and FP states are larger than in the NP state. It may be due to of the increased laser energy input; the keyhole (see Figure 5) and the metal vapor ejections are becoming more intense, exhibiting much stronger fluctuations (larger amplitude and frequency). While the NP state does not have a deep enough keyhole, the laser energy is relatively small, melting the metal more slowly, making the metal vapor and plasma ejection less intense and the keyhole depression slower, so the AE event occurs in the low-frequency range. 

Therefore, the results of spectrum distribution further indicate that the keyhole and its surrounding liquid molten pool layer act as a frequency-selective amplifier for pressure fluctuations induced by changes in the interaction of pulsed laser radiation with the keyhole wall. The experimental data show a strong correlation between the acoustic emission spectrum at 1~10 kHz and keyhole behavior as well as weld penetration. These acoustic spectra undoubtedly reflect a variety of quasi-periodic phenomena that are characteristic of the laser-metal interaction during laser welding. It is worth noting that an individual frequency component at a given frequency has no clear physical significance, which is not sufficient to be used as an efficient acoustic feature.

## 3. VMD-Based Frequency Component Extraction

To investigate more about the feasibility of distinguishing between low-frequency and high-frequency components for better describing the weld penetration, an efficient frequency decomposing technique is crucial. In this research, we apply the VMD (Variational Mode Decomposition) method, which is an adaptive signal processing method that decomposes a none-stationary AE signal *f* into *K* modal component sub-signals *u_k_* with each component having a defined finite bandwidth and a central frequency *ω_k_*. The method assumes that any signal consists of a series of sub-signals with a specific center frequency and finite bandwidth, i.e., an IMF (intrinsic mode function) [34].

Based on the classical Wiener filter, the modal function is obtained by solving the variational problem to obtain the central frequencies and bandwidth limits and finding the effective components corresponding to each central frequency in the frequency domain. The variational problem for the constraint is:(9)$$\begin{array}{c} \left\{\begin{array}{c} \underset{\begin{array}{cc} \left\{u_{k}\right\} & \left\{\omega_{k}\right\} \end{array}}{\min}\left\{\overset{K}{\underset{k=1}{\sum }}||\partial_{t}\left[\left(\delta \left(t\right)+\frac{j}{\pi t}\right)*u_{k}\left(t\right)\right]e^{-j\omega_{k}t}|{|}_{2}^{2}\right\}\\ s.t.\overset{K}{\underset{k=1}{\sum }}u_{k}=f \end{array}\right\} \end{array}$$
where: {*u_k_*} = {*u*_1_,…, *u_K_*}, {*ω_k_*} = {*ω*_1_,…, *ω_K_*} are all modes and their central frequencies, respectively; *∂_t_* is the partial derivative with respect to *t*, *δ*(*t*) is the Dirac distribution, and * denotes the convolution. 

To solve Equation (9), a quadratic penalty factor *α* and a Lagrange multiplier operator *λ* can be introduced to convert Equation (9) into an unconstrained variational problem, the augmented Lagrangian function is expressed as in Equation (10):(10)$$\begin{array}{l} L\left(\left\{u_{k}\right\},\left\{\omega_{k}\right\},\lambda \right)=\alpha \overset{K}{\underset{k=1}{\sum }}||\partial_{t}\left[\left(\delta \left(t\right)+\frac{j}{\pi t}\right)*u_{k}\left(t\right)\right]e^{-i\omega_{k}t}|{|}_{2}^{2}+||f\left(t\right)-\overset{K}{\underset{k=1}{\sum }}u_{k}\left(t\right)|{|}_{2}^{2}\\ \;\;\;\;\;\;\;+\langle \lambda \left(t\right),f\left(t\right)-\overset{K}{\underset{k=1}{\sum }}u_{k}\left(t\right)\rangle \end{array}$$

By using the alternating directional multiplier (ADMM) iterative algorithm combined with the Parseval/Plancherel, Fourier isometric transform, the modal components, and central frequencies were all optimally obtained, and the saddle points of the extended Lagrange function were searched for. The expressions for *u_k_*, *ω_k,_* and *λ* after the alternating search iteration are as follows [34]:(11)$${\hat{u}}_{k}^{n+1}\left(\omega \right)=\frac{\hat{f}\left(\omega \right)-\sum_{i\neq k}{\hat{u}}_{i}\left(\omega \right)+\frac{\hat{\lambda }\left(\omega \right)}{2}}{1+2\alpha {\left(\omega -\omega_{k}\right)}^{2}}$$

The same method of updating the central frequency is understood to be:(12)$$\begin{array}{c} \omega_{k}^{n+1}=\frac{\int_{0}^{\infty }\omega {\left|{\hat{u}}_{k}\left(\omega \right)\right|}^{2}d\omega }{\int_{0}^{\infty }{\left|{\hat{u}}_{k}\left(\omega \right)\right|}^{2}d\omega } \end{array}$$
where ${\hat{\mu }}_{k}^{n+1}\left(\omega \right)\;$ is equivalent to the Wiener filter of the current residual $\hat{f}\left(\omega \right)-\sum_{i\neq k}{\hat{u}}_{i}\left(\omega \right)$; $\omega_{k}^{n+1}$ is the center of gravity of the current eigenmode power spectrum.

When using VMD for signal processing, a *K* value that is too large or too small will result in spurious or missing components, so the parameter *K* needs to be optimized. Using the instantaneous frequency averaging method [35], the instantaneous frequency of the *j*th sample point is obtained as *f_ij_*, assuming that the *i*th modal component of the original signal has *M* samples. The original signal was pre-decomposed using the VMD method, and the modal classification at *K* = 2~9 was Hilbert transformed to obtain the resolved signals. By applying the instantaneous frequency averaging method:(13)$$f_{i}=\frac{1}{N}\underset{j}{\sum }f_{ij}$$

Where *N* is the number of instantaneous frequencies of a modal component, and the penalty factor *α* was set to 2000. The normalized instantaneous frequency averaging curves for VMD modal components 2 to 9 were analyzed, as depicted in Figure 5.

It can be seen that when the modal component increases to a certain value, the curve bends distinctly from the neighboring points, and the instantaneous frequency mean of the modal component at that point changes abruptly, indicating that the different sub-signals produce abrupt changes in frequency during the decomposition. If the curve does not bend significantly but forms a gentler horizontal or vertical line with the adjacent points, it indicates that the different sub-signals have generated frequency blending during the decomposition of the signal. According to the instantaneous frequency averaging transformation curves from *K*2 to *K*9 in Figure 6, the result of the VMD decomposition layer at *K*9 is a relatively smooth folded line. There is no sudden change in the frequency components of each mode, which means that the VMD method at *K*9 can better decompose the raw AE signals into different IMF components.

Taking a full-penetration AE signal as an example, the signal was decomposed into 9-layer IMF components using the VMD method, and the penalty factor *α* was set to 2000 based on the discrimination method of center frequency observation. Figure 7a illustrates the various reconstitution sub-signals of each layer of the IMF. Then the FFT-based spectral analysis was performed for each IMF layer to visualize the nine IMF components of the raw AE signal, as shown in Figure 7b.

It can be seen that the frequency distribution of the IMF sub-signal derived from the raw AE signal after 9-layer decomposition is more reasonable, which is consistent with the interpretation of the instantaneous frequency averaging method. To calculate whether the VMD method separates signals with high energy in the low-frequency band, the time domain energy calculation method is introduced:(14)$$\begin{array}{c} En\left(x\right)=\int_{0}^{N}|x\left(t\right){|}^{2}dt \end{array}$$

*N* is the maximum number of time series; *N* is set to 800,000. The energy percentage of each sub-signal is calculated as shown in Figure 8. It indicates that the 9th layer IMF component is in the lowest frequency band with the highest energy, and the remaining 8 IMF components are in higher frequency bands with lower energy. The comparison result demonstrates the feasibility and veracity of the VMD technique in reflecting the acoustic energy distribution among the low-frequency and high-frequency components of AE signals. 

## 4. The Proposed Spatio-Temporal CNN-LSTM Model 

During the pulsed laser welding process, affected by the high transient characteristics of the laser beam, the AE signal presents a highly dynamic, complex, and changeable characteristic. In particular, the energy distribution of different components in AE signals over time would vary under different welding conditions based on VMD pre-processing. Therefore, a deep learning algorithm based on the fusion of spatio-temporal information was proposed to make full use of the time-dependent relationship and spatial characteristics of the time-frequency information in a time-varying AE signal. In this section, a 1-D CNN model was firstly applied to deeply explore the spatial feature differences between the low-frequency high-energy components and the high-frequency low-energy components. After extracting the spatial features of the AE signal, an LSTM model was connected to mine the temporal features of this time series. Then, a hybrid CNN-LSTM model was constructed to predict the laser welding penetration status.

### 4.1. CNN Framework

In this study, the 1-D CNN architecture was used, and the decomposed AE sub-signals using VMD were used as the model input. Due to the advantage that 1-D CNN only convolves the filter width without the filter height [36], it can effectively convolve the 9-layer sub-signals at each time-series sample to deeply explore the energy distribution feature between low-frequency and high-frequency components. As shown in Figure 9, the filter width and height were set to 9 and 1, and the width and height of each sequence vector were 9 and 480,000. The filter sliding step was 1 from the start position along the height direction sliding to the final position, for a total of 479,999 sliding times. The second sequence vector was obtained after one filter convolution operation with the width and height changing, and was calculated as follows:(15)$$\begin{array}{c} H_{2}=\frac{H_{1}-F_{H1}}{S_{1}}+1 \end{array}$$
(16)$$\begin{array}{c} W_{2}=\frac{W_{1}-F_{W1}}{S_{1}}+1 \end{array}$$

Taking the height calculation as an example, *H*_1_ and *H*_2_ are the heights of the first and second sequence vectors, respectively; *F_H_*_1_ is the height of the first convolutional layer; and *S*_1_ is the sliding length. The width was also calculated in the same way. Two layers of 1-D CNN were used to extract the sequence vector features in turn. Nine filters were set in the first layer to extract the width features of the sequence vectors, resulting in sequence vectors of 9 × 480,000. The second layer was set with six filters to extract the width features of the sequence vectors, resulting in sequence vectors of 6 × 480,000. Each 1-D CNN layer was then connected with one RELU layer as an excitation function, and a dropout layer was used at the end of the two convolution layers to prevent over-fitting. 

### 4.2. LSTM Framework

LSTM (Long Short Term Memory) has a more accurate information transfer mechanism and can automatically store and remove temporal state information compared to RNN (Recurrent Neural Network). In addition, it can effectively solve the gradient disappearance problem and dependency problems caused by long input sequences [37]. As shown in Figure 10, the basic structure of the LSTM model uses input gates, output gates, and forgetting gates to protect and control the state of the storage unit. 

Firstly, the input *X_t_* is used in conjunction with the previous output *h_t_*_−1_, in order to obtain the output *f_t_* of the forgetting gate; the output range is used to filter whether the output of the previous unit is remembered, or not. The input gate is then responsible for selectively remembering the current input information and outputting it, and together they update the state information in the long-term memory chain *C_t_*. Finally, *h_t_*_−1_ and *X_t_* are passed through the output gate and multiplied by the activation function to obtain the output *h_t_* for that cell. The flow of information in the LSTM can be described as the following equation:(17)$$\begin{array}{c} f_{t}=\sigma \cdot \left(W_{f}\cdot \left[h_{t-1}+X_{t}\right]+b_{t}\right) \end{array}$$
(18)$$\begin{array}{c} i_{t}=\sigma \cdot (W_{i}\cdot \left[h_{t-1}+X_{t}\right]+b_{i}) \end{array}$$
(19)$$\begin{array}{c} C_{t}^{\prime }=tanh\cdot \left(W_{c}\cdot \left[h_{t-1}+X_{t}\right]+b_{C}\right) \end{array}$$
(20)$$\begin{array}{c} C_{t}=f_{t}*C_{t-1}+i_{t}*{C^{\prime }}_{t} \end{array}$$
(21)$$\begin{array}{c} o_{t}=\sigma \cdot (W_{o}\cdot \left[h_{t-1}+X_{t}\right]+b_{o}) \end{array}$$
(22)$$\begin{array}{c} h_{t}=o_{t}*tanh\left(C_{t}\right) \end{array}$$
where *W_f_*, *W_i_*, *W_o_*, *W_c_* are weighting coefficients and *b_t_*, *b_i_*, *b_c_* are bias coefficients.

After extracting the deep spatial features of the temporal vectors using two 1-D CNN layers, the time-series size of the temporal vector became 6 × 480,000. Then, a LSTM layer was connected to explore the temporal features of this time series. As shown in Figure 11, one pulse period of the pulsed laser is 10 ms, and the sampling frequency was set to 20 kHz, i.e., one pulse period covers 2000 sampling points. With one pulse cycle as one frame, there are 240 frames within 480,000 sequence points. Therefore, the number of hidden units in the LSTM layer was fixed to 240, and each hidden unit processed one pulse cycle, which greatly speeds up the efficiency of the LSTM layer. 

### 4.3. Establishment of the CNN-LSTM Hybrid Model

After adaptively decomposing the raw AE signal into 9-layer sub-signals with the VMD technique, a two-layer 1-D CNN model fully extracts the spatial features characterizing the time-frequency energy distribution of each sub-signal, while the LSTM model extracts the temporal state information reflecting the keyhole dynamics. Thus, the combination of CNN-LSTM hybrid models can describe the complex variation trend of acoustic waves under different welding conditions and obtain more accurate penetration predicting results. After combining with CNN and LSTM layers, a dropout layer was connected to prevent over-fitting with a dropout value of 0.5, followed by a FC (fully connected) layer, a Softmax activation function layer, and a classification layer including three types of penetration states (NP, PP, and FP). Figure 12 depicts the basic architecture of the CNN-LSTM hybrid model, which will be trained in order to verify the model’s performance in the next section. 

## 5. Results and Discussion

### 5.1. Performance Evaluation of the Constructed Model

In this subsection, the predicting accuracy of different penetration states was evaluated using the constructed deep learning model. Due to the large number of parameters involved in CNN-LSTM, many hyperparameters (e.g., number of CNN or LSTM layers, batch size, and CNN filter number) directly affect the model recognition accuracy. Based on a great deal of experimentation, the optimized hyperparameters are provided in Figure 13; the rest of the parameters are defaults. A total of 800,000 samples of data were available for each of the three penetration states. 60% of the sample data from each state was used as the training set, 20% as the validation set, and the remaining 20% as the testing set. All experiments were deployed based on the Matlab 2022a environment and validated, using a computer equipped with an Intel(R) core(TM) i7-10750H processor and a NVIDIA RTX2060 GPU. The performances of implemented methods were assessed and compared within terms of recognition, accuracy, and corresponding loss values.

According to Figure 13, although there is a decline in accuracy after 40 iterations, the proposed model has a higher overall recognition accuracy of 99.8% after 150 training iterations. Meanwhile, the training and validation of model loss converged to zero after completing iterating. In addition, the normalized confusion matrix was calculated to clearly display the classification and misclassification results of each penetration status under different feature representations. As depicted in Figure 14, only one sample was misclassified as an FP state in an NP state classification, and the recognition accuracy reached about 100%. In the PP state classification, 183 samples were misclassified as NP, and the recognition accuracy reached 99.9%. In the FP state classification, 539 samples were misclassified as PP state, and the recognition accuracy reached 99.7%. The CNN-LSTM hybrid model can obtain a very high classification accuracy of 99.8%. 

### 5.2. Performance Comparison between Different Models

To further demonstrate the superiority of the CNN-LSTM model, the performance comparison with other models was verified by examining the test accuracy. The effects of different network structures on the model’s performance were also tested. To guarantee the reliability of the proposed approach, 5-fold cross-validation was applied, and the average of those five times was calculated as the final accuracy of each model. The standard deviation (STD) was also calculated for each model’s accuracy to judge the model’s stability. The performance statistics of the replicate experiments for each model were depicted and compared in Figure 15, the parameters of the different networks are detailed in Table 3, and the following conclusions can be drawn:No. 1 and No. 3 network models only compared the effect of different hidden layer nodes in LSTM on the model performance when other parameters remained unchanged. As the number of hidden nodes increases from 120 to 240, the average recognition accuracy also increases from 96.43% to 98.50%, which indicates that a lager node number in the LSTM hidden layer has a higher classification performance;Since the 1st layer of CNN was applied to extract the spatial features of 9-layer sub-signals through VMD, the filter number of the 1st layer was fixed at 9, thus No. 2–No. 5 network models compared the effect of the filter number of the 2nd CNN layer in turn. It is worth noting that the No. 4 model has a best classification performance of 99.85% when the filter number of 2nd layer is set to 6. Meanwhile, it has a minimum accuracy fluctuation when STD = 0.21, confirming that the optimal model is stable. Although other models (No. 2 and No. 5) can also reach a larger recognition accuracy exceeding 98%, there exists a strong accuracy fluctuation at 50 or 100 iterations, indicating that these models are not stable;As shown in Figure 15b, the No. 4 and No. 6 compared the effect of the VMD-based frequency components on the model’s performance. It can be found that the input data of the No. 6 network was not decomposed using the VMD method, which led to a significant accuracy (67.83%) decrease compared to the No. 4 network with the VMD method;Lastly, the No. 4 and No. 7 compared the effect of the CNN-based spatial feature extraction on the model performance after the VMD processing. The final recognition classification of No. 4 (99.85%) is higher than No. 7 (95.04%), and the standard deviation of No. 4 (0.21) is also lower compared to No. 7 (0.28). Meanwhile, the No. 4 structure achieves the global optimum in fewer iterations, indicating that the CNN layer contributes greatly to the penetration and prediction of results.

**Table 3 materials-16-01614-t003:** Detailed hyperparameters and performance statistics for different networks.

No.	1	2	3	4	5	6	7
VMD method	Yes	Yes	Yes	**Yes**	Yes	No	Yes
Filter number of 1st CNN layer	9	9	9	**9**	9	-	-
Filter number of 2nd CNN layer	3	1	3	**6**	9	-	-
LSTM hidden nodes	120	240	240	**240**	240	240	240
Max Acc(%)	97.45	98.50	98.82	**99.99**	99.99	70.32	96.25
Min Acc(%)	96.34	97.78	98.10	**99.31**	97.31	65.33	94.16
Avg Acc(%)	96.43	98.16	98.50	**99.85**	98.37	67.83	95.04
Standard deviation	0.28	2.33	0.25	**0.21**	2.32	0.32	0.28

**Figure 15 materials-16-01614-f015:**
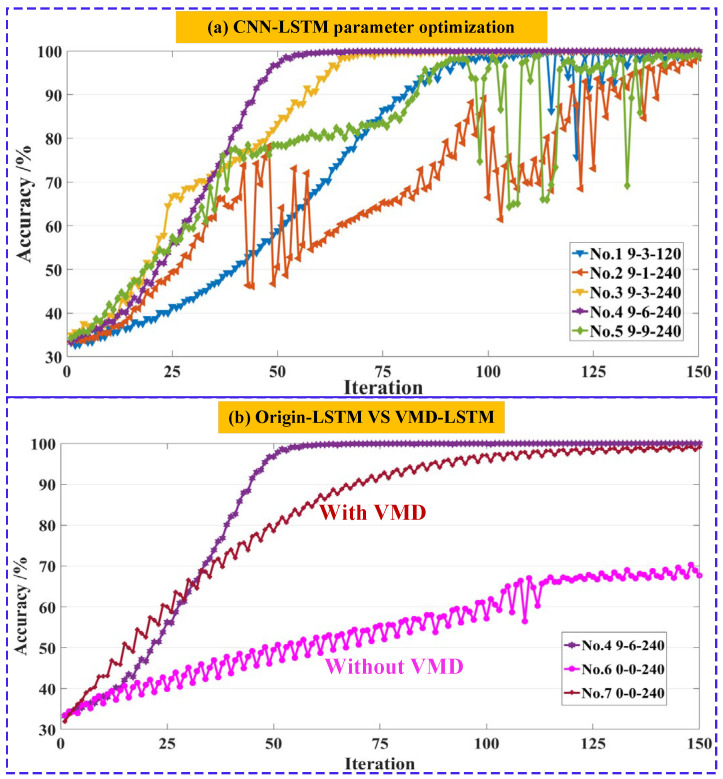
(**a**) Comparison results between No. 1~No. 5 network models. For instance, the network structure of 9-3-120 means it has 9 filters in the 1st layer, 3 filters in the 2nd layer of the CNN model, 120 hidden nodes in the LSTM layer, and so on. (**b**) Comparison results between No. 4, 6, and 7 network models. No. 6 and No. 7 networks reflect the model only having a LSTM layer without a CNN layer, namely Origin-LSTM. No. 6 network deals with the raw AE signals without VMD, while the No. 7 network deals with the decomposed AE signals with VMD.

In conclusion, the combination of the detected AE signals, VMD processing, and CNN-LSTM approach (No. 4 network) can obviously contribute to achieving excellent performance for predicting the penetration status during pulsed laser welding. The reason can be explained: the VMD technique can efficiently decompose the non-stationary AE signal into low-frequency and high-frequency components. After that, the proposed approach integrates the excellent spatio-temporal feature extraction capabilities of CNN and LSTM for dealing with the extracted frequency components. Compared to traditional feature extraction and machine learning methods used in other studies, such as in [15,16], the present method does not require the reduction of noise collected by an acoustic emission sensor and exhibits better adaptive feature extraction properties as well as higher prediction accuracy and a lower standard deviation, which is undoubtedly a great improvement in the model. Thus, the hybrid model provides a more accurate dynamic modeling approach for monitoring welding penetration based on AE sensing while being able to adequately characterize the keyhole dynamics during pulsed laser welding.

## 6. Conclusions and Future Works

This work developed a novel acoustic in-process penetration monitoring method based on the acoustic wave characteristics of a CNN-LSTM hybrid model for aluminum alloys in pulsed laser welding. The following conclusions can be drawn: Combining with the high-speed photography and AE measurements, the characteristics of the AE signal are closely related to the keyhole oscillation and various weld penetrations. Based on the proposed mechanism of an acoustic source, the keyhole oscillation under the dynamic pressure fluctuation is considered a potential point acoustic source, and the surrounding molten pool acts like a speaker diaphragm, generating the spherical acoustic waves propagating in the workpiece;According to the STFT time-frequency analysis, the acoustic spectra undoubtedly reflect a variety of quasi-periodic phenomena that are characteristic of the laser-metal interaction during laser welding; then the proposed VMD technique adaptively decomposed the raw AE signal into nine distinct frequency components, which can precisely characterize the acoustic energy distribution among the low-frequency and high-frequency components, under different welding penetrations, and improved frequency domain identifiability;Finally, a novel hybrid model combing CNN and LSTM was designed to deeply mine the spatial and temporal acoustic features from the extracted frequency components. Extensive experiments demonstrate that our proposed approach yields a remarkable classification performance with a test accuracy of 99.8% and a standard deviation of 0.21, which obtains the best recognition performance compared with other deep learning methods.

The current research focuses mainly on the penetration status classification for the fully penetrated butt-joint welds during pulsed laser welding. Aiming at non-penetrated welds (lap or fillet joints), our future study will apply the acoustic emission and CNN-LSTM techniques for penetration depth regression forecasting, which can further verify its reliability and broaden its industrial application domain. In addition, future works will also focus on the combination of acoustic emission with thermal and photoelectric sensors and give full play to the potential advantages among monitoring techniques. It will provide the possibility to detect internal macroscopic seam defects (cracks and pores) and deliver better performance prediction during highly dynamic laser welding.

## Figures and Tables

**Figure 1 materials-16-01614-f001:**
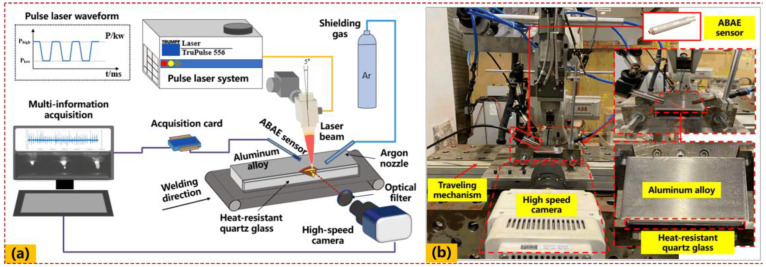
View of the pulsed laser welding experimental platform and multi-information acquisition system: (**a**) schematic diagram; (**b**) experimental configuration.

**Figure 2 materials-16-01614-f002:**
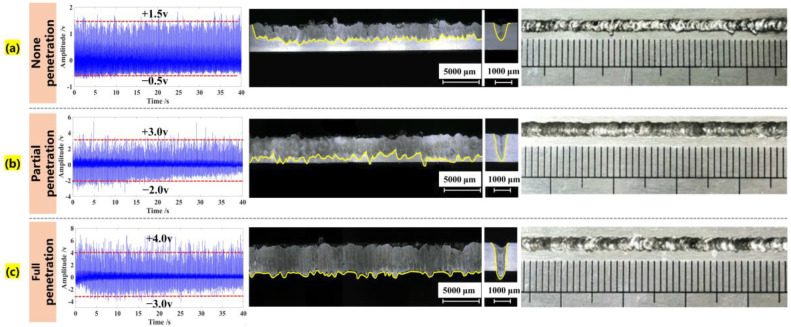
Weld cross-section and top morphology as well as AE signals under different weld penetrations.

**Figure 3 materials-16-01614-f003:**
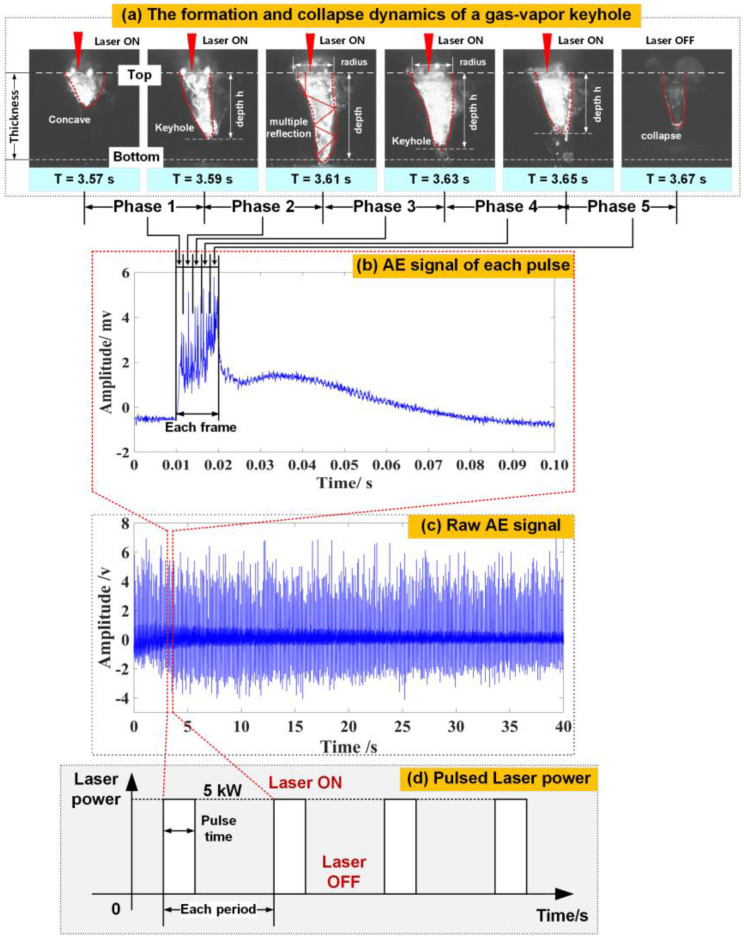
Relationship between the keyhole behavior and its AE signal under full-penetration state during the repetitively pulsed laser welding: (**a**) formation and collapse dynamics of a gas-vapor keyhole; (**b**) waveform of a single pulsed AE event; (**c**) waveform of a raw AE signal; (**d**) time-dependent distribution of pulsed laser power.

**Figure 4 materials-16-01614-f004:**
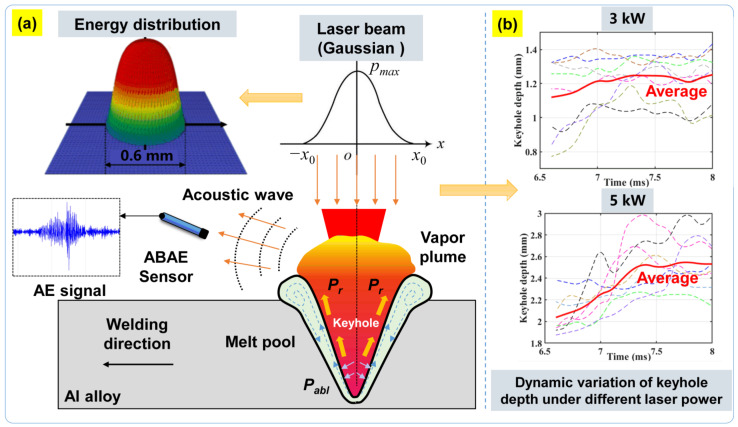
Generation mechanism of acoustic waves during the pulsed laser welding process: (**a**) the pressure balance inside the keyhole under the laser beam radiation; (**b**) dynamic variation of keyhole depth under different laser powers.

**Figure 5 materials-16-01614-f005:**
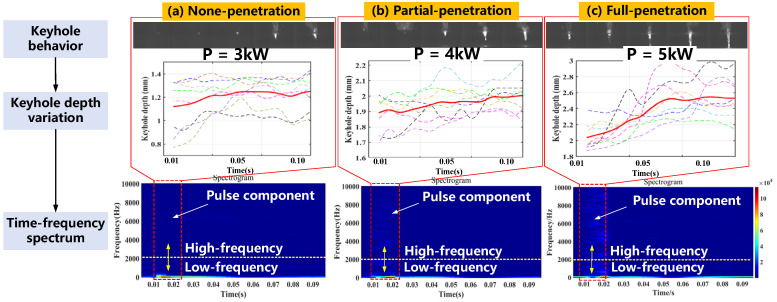
Comparison of time-frequency characteristics with a 2-D spectrogram and keyhole depth variation under different weld penetrations.

**Figure 6 materials-16-01614-f006:**
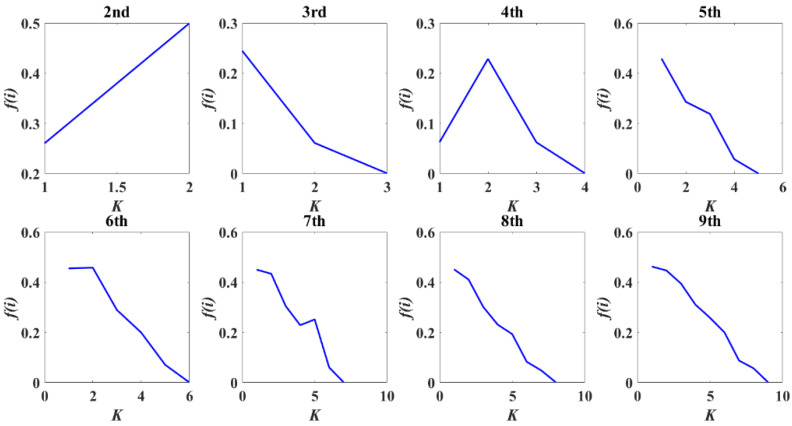
Standardized instantaneous frequency averaging curves of each modal from decomposition layer *K*2 to *K*9.

**Figure 7 materials-16-01614-f007:**
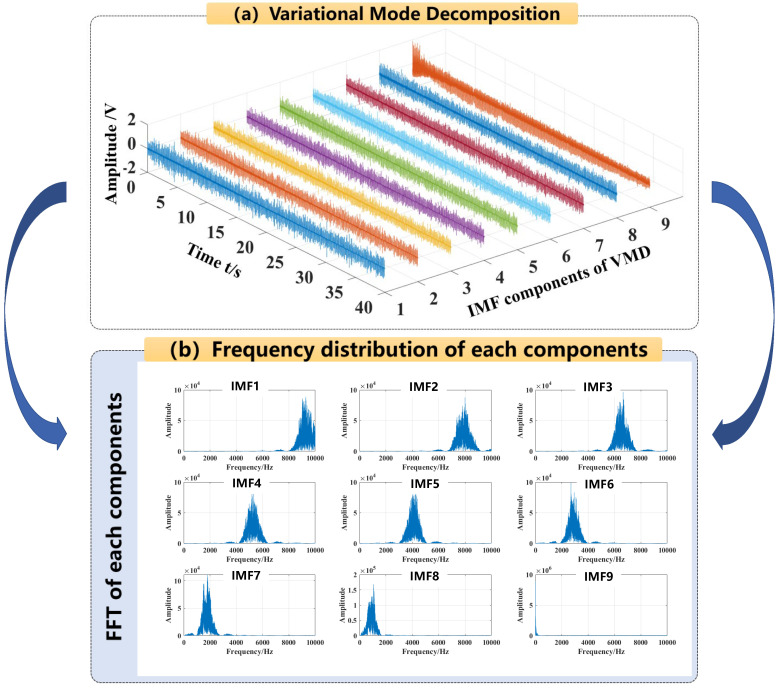
9-layer IMF component decomposition using VMD and FFT-based spectral visualization. (**a**) shows the variational mode decomposition of the original signal; (**b**) shows the frequency distribution of each components according to the original signal.

**Figure 8 materials-16-01614-f008:**
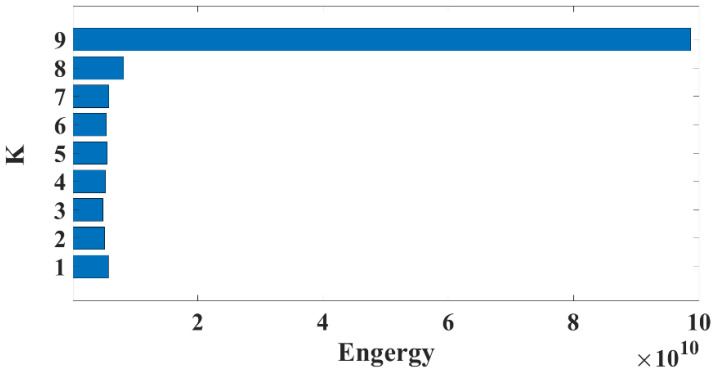
Energy percentage of each IMF sub-signal using the VMD method.

**Figure 9 materials-16-01614-f009:**
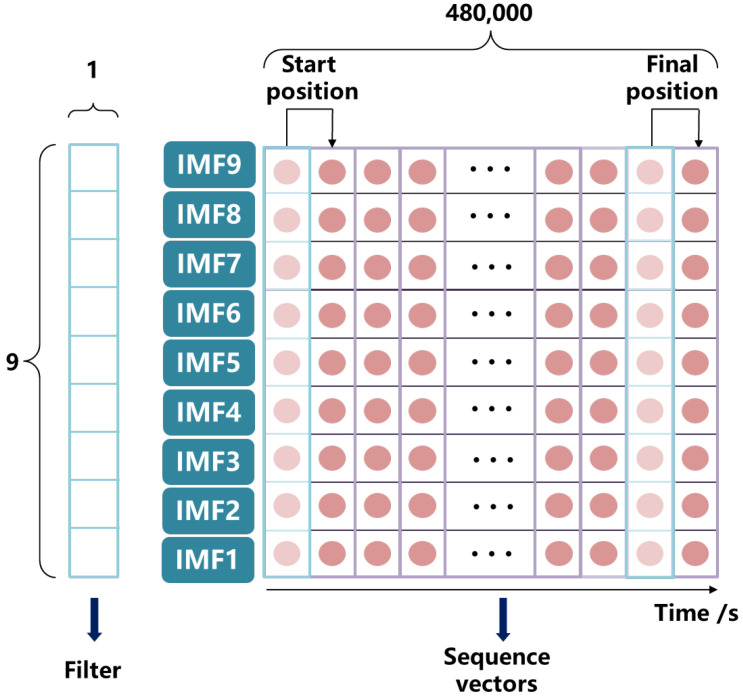
1-D Convolutional Neural Network structure for deeply exploring the energy distribution feature between low-frequency and high-frequency components (Structure includes 1-D convolution kernel, i.e., filter, and sequence vectors. Width × Height of the filter and sequence vectors are 9 × 1 and 9 × 480,000 respectively).

**Figure 10 materials-16-01614-f010:**
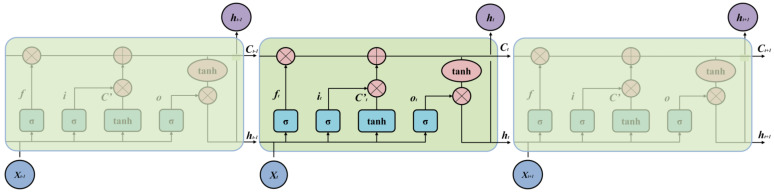
The detailed architecture of the LSTM model.

**Figure 11 materials-16-01614-f011:**
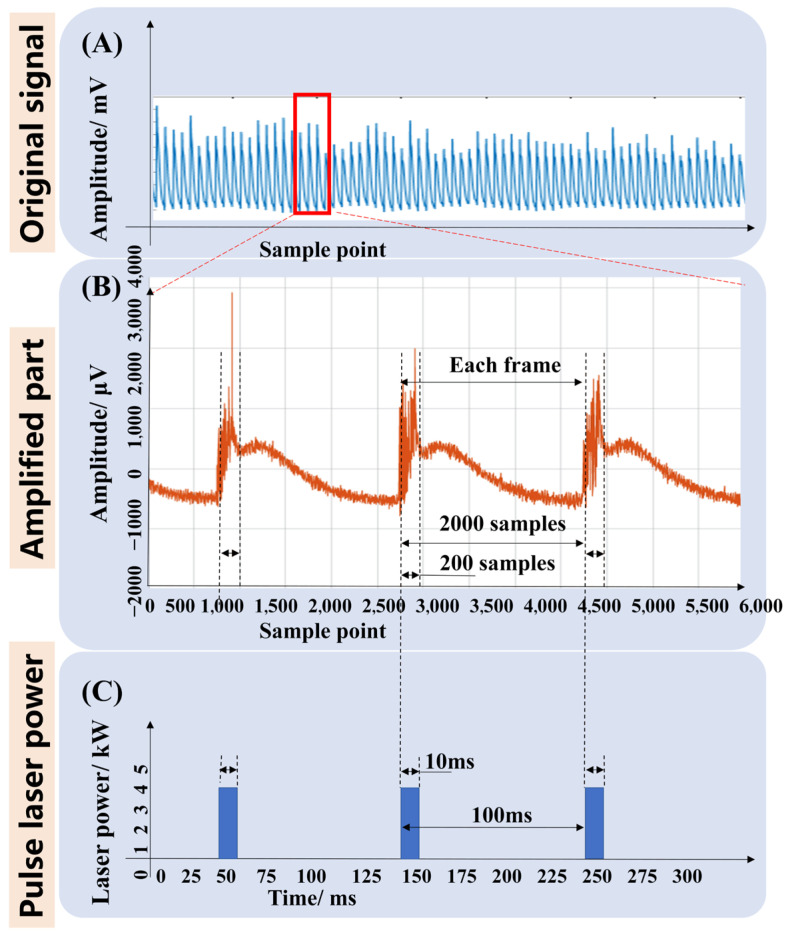
The time-series sample determination for the LSTM model.((**A**) shows the orininal signal of the ABAE sensor; (**B**) shows the amplified part of the red box in (**A**); (**C**) shows the power of the pulsed laser machine mode.).

**Figure 12 materials-16-01614-f012:**
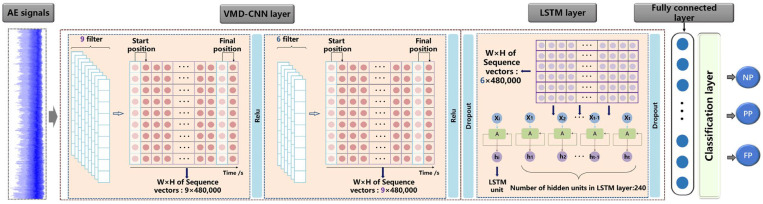
The basic architecture of the CNN-LSTM hybrid model.

**Figure 13 materials-16-01614-f013:**
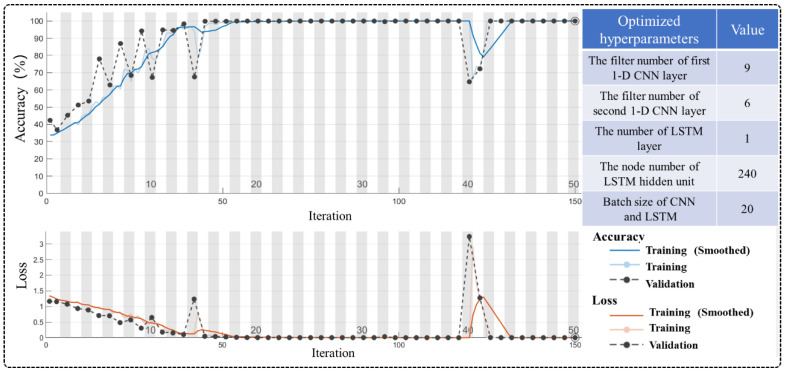
Optimized hyperparameters of the proposed CNN-LSTM model and the final training/validation recognition accuracy after 150 iterations.

**Figure 14 materials-16-01614-f014:**
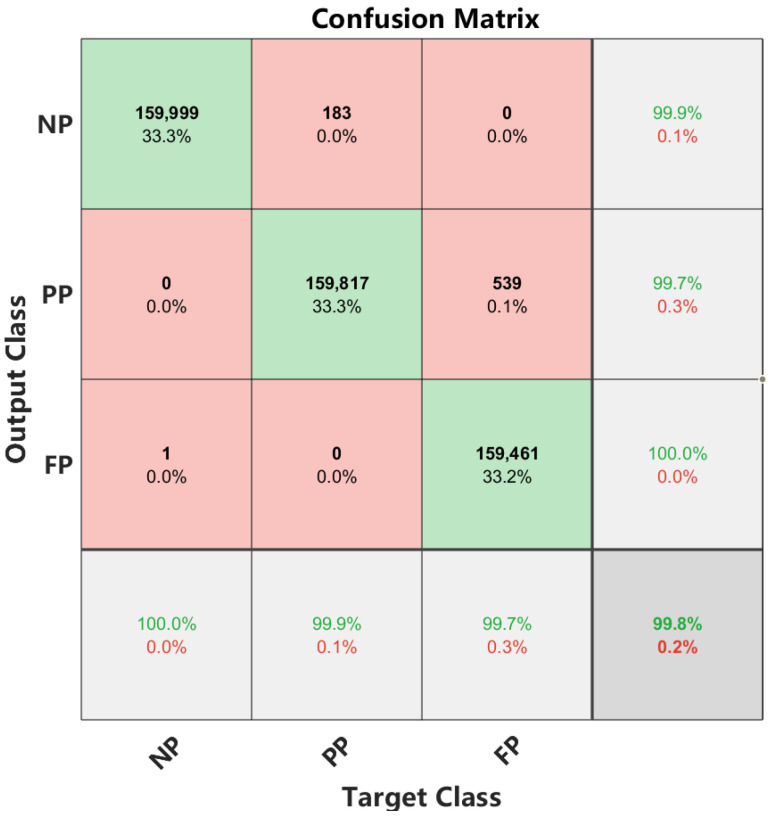
Confusion matrix of the VMD-CNN-LSTM model.

**Table 1 materials-16-01614-t001:** Detailed welding parameters used in experiments and corresponding welding quality.

Case	Laser Power	Pulse Width	Pulse Frequency	Welding Speed	Defocusing Distance	Penetration Status
1	5 kW	10 ms	10 Hz	20 mm/s	0 mm	FP
2	4 kW	10 ms	10 Hz	20 mm/s	0 mm	PP
3	3 kW	10 ms	10 Hz	20 mm/s	0 mm	NP

(Full-penetration: FP, Partial-penetration: PP; None-penetration: NP).

**Table 2 materials-16-01614-t002:** The national standards for the chemical composition and the mechanical properties of 6061 alloys.

Alloy	Si (%)	Fe (%)	Cu (%)	Mn (%)	Mg (%)	Cr (%)	Zn (%)	Ti (%)	Tensile Strength (MPa)	Yield Strength (MPa)	Elongation after Fracture (%)
6061	0.4–0.8	0.7	0.15–0.4	0.15	0.8–1.20	0.04–0.35	0.25	0.15	≥290	≥240	7

## Data Availability

The raw/processed data and modeling codes required to reproduce these findings cannot be shared at this time as the data also forms part of an ongoing study.

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
