# Peer review of "Laser Welding Penetration Monitoring Based on Time-Frequency Characterization of Acoustic Emission and CNN-LSTM Hybrid Network"

_materials, 2023, doi:10.3390/ma16041614_

Round 1

Reviewer 1 Report

The manuscript entitled ” Laser welding penetration monitoring based on time-frequency characterization of acoustic emission and CNN-LSTM hybrid network” by Luo et al. is interesting, while combining laser welding, acoustic emission and deep learning, provides new information and is suitable in Materials after a minor revision.

Suggestions:

-          General: Why are the links to the references written as X and not [X], where X is a reference number? Please update everywhere.

-          General: Please use upright fonts in the subscripts if they are abbreviations, like “abl”. If they are variables, like “k” then keep the italic fonts.

-          P15, L496: Change “sequences35” to “sequences [35]”

-          P15, eqts (19) and (22): “Tanh” should be in upright fonts – not in italics as it is an operator

-          P17: Please enlarge Figure 12 – it is difficult to see the details

-          P21-22, References: Please follow the author guidelines and be consistent with the references. For example, the journal names should be either fully spelled or abbreviated. Not mixed as it is now.

-          P22, L728: Change “Neural computation” to “Neural Computation” or its abbreviation

Author Response

Point 1:

 General: Why are the links to the references written as X and not [X], where X is a reference number? Please update everywhere.

General: Please use upright fonts in the subscripts if they are abbreviations, like “abl”. If they are variables, like “k” then keep the italic fonts.

Response 1: 

Thank you for your question and we regret to say that the formatting issues based on the references were all due to machine typographical errors. We apologize for any inconvenience caused and have made formatting changes in the manuscript, as well as carefully checking the abbreviations throughout the text and improving them.

Point 2:

P15, L496: Change “sequences35” to “sequences [35]”

P15, eqts (19) and (22): “Tanh” should be in upright fonts – not in italics as it is an operator

P17: Please enlarge Figure 12 – it is difficult to see the details

P21-22, References: Please follow the author guidelines and be consistent with the references. For example, the journal names should be either fully spelled or abbreviated. Not mixed as it is now.

          P22, L728: Change “Neural computation” to “Neural Computation” or its abbreviation

Response 2: 

Thank you for your suggestion, we have accepted your suggestion in P15, L496, eqts (19) and eqts (22) and made the changes. We have also enlarged the text in Figure 12 appropriately for viewing purposes.

All the corrections please see our new manuscript in the attachment.

Reviewer 2 Report

Dear Authors,  

I have reviewed your paper titled: 

Laser welding penetration monitoring based on time-frequency characterization of acoustic emission and CNN-LSTM hybrid networ.

The paper fulfils the aims and scope of Materials journal, and can be considered for potential publication. I have some suggestions, which are listed below. 

General remarks:

- Please add the quantitative results into the abstract.
- All equations should be supported with relevant reference(s).
- The style of citations is wrong and out of template. References should be cited in bracket [ ].

Introduction:

- This part is very short. Moreover, it is not supported with any references.

Acoustic emission monitoring:

- This section is numbered as "2". There is the second section numbered as "2". Moroever, in my opinion, this section and section " Machine learning modeling" (numbered as 3, the same as other section) are clearly part of introduction, and should be presented as one section.

- You have not presented any issues about welding of used base material. Yopu should show, which methods could be used, which problems could be observed. It should underline, that your method could be used as the best.

- The novelty of your work should be underlined well. Now, it is not clear, what new has been proposed in this paper.

Methodical approach:

- You have not presented: chemical composition, mechanical properties of investiated material.

- Table 1 - why these parameters were used? It is not clear. Potential readers should have clear information from the text.

Results and discussions:

- This section requires modifications. Firsly, there is no proof that you have welded any samples. Please show some photos. Moreover, photos will proove that used welding parameters were good, and allow to create full joint.

- I cannot find any scientific discussion here. Paper looks more like technical raport than scientific paper. You should compare your results with other scientific papers. It allows to underline the biggest advantages from your work. Moreover, it will proov the necessity and novelty of your work.

Conclusions:

- Please support with the quantitative results.

Author Response

Point 1:

General remarks:

1.Please add the quantitative results into the abstract.

2.All equations should be supported with relevant reference(s).

3.The style of citations is wrong and out of template. References should be cited in bracket [ ].

Response 1:

Thanks to your suggestion, we have put more emphasis in the abstract on the high accuracy results of the models constructed in this paper, which is necessary for the article.

Based on question 2, there are some equations that are well known in the industry, such as the Fourier transform equation and the short-time Fourier transform equation, and some equations that are derivations of the CNN and the LSTM and are also official equations, which are written at the bottom of the model introduction and therefore do not identify the reference source.

Of course for Eqs. 1-6 of the original acoustic source theory in our text, we have made corrections to add theoretical reference sources.

Based on question 3, we have corrected the formatting and we need to clarify that the formatting of the references is due to a machine formatting error and differs from the layout of the original PDF, so we apologise for any inconvenience caused.

Point 2:

Introduction:

This part is very short. Moreover, it is not supported with any references.

Acoustic emission monitoring:

- This section is numbered as "2". There is the second section numbered as "2". Moroever, in my opinion, this section and section" Machine learning modeling" (numbered as 3, the same as other section) are clearly part of introduction, and should be presented as one section.

Response 2:

Thank you for your suggestion, we have made a correction, the corrected presentation includes a section on acoustic emission monitoring and a section on machine learning, this is also due to a machine formatting error, please accept our apologies for any inconvenience caused.

Point 3:

- You have not presented any issues about welding of used base material. Yopu should show, which methods could be used, which problems could be observed. It should underline, that your method could be used as the best.

Response 3:

Thanks for your question.

There are many sensing methods in the study of laser welding process monitoring, which leads to process monitoring being a broad area of research, and we therefore describe in our introduction the use of optical, thermosensitive sensing means for laser welding process monitoring.

In section 2.3 we describe the mechanism of keyhole acoustic generation, describe how acoustic waves that can be collected by acoustic emission sensing are generated and thus describe the need to use acoustic emission for monitoring. Finally we describe the principles and results of the method used in this paper in Sections 3 and 4 and demonstrate in Table 2 in Section 5 that the proposed method achieves the best results.

Point 4:

- The novelty of your work should be underlined well. Now, it is not clear, what new has been proposed in this paper.

Response 4:

Thanks to your suggestions, we agree that scientific research on novelty is essential and that this is the main thrust of this paper. In a brief summary, in order to extract information about the large welding features contained in the acoustic waves generated by the keyhole, a VMD-CNN-LSTM method is used.

The VMD approach decomposes the acoustic signal into nine different central frequencies, which increase the frequency domain discriminability of the acoustic information.

The 1D CNN method extracts the spatial features of the ascending acoustic wave information, and the LSTM method extracts the temporal features of the acoustic wave information, and the three methods are combined to construct a spatial-temporal neural network architecture, and after a large amount of data training, the information in the original 1D acoustic wave that can characterize the penetration state can be easily identified, and then the model shows excellent recognition rate. These are the novelties of this work.

Point 5:

Methodical approach:

- You have not presented: chemical composition, mechanical properties of investiated material.

Response 5:

Thank you for your question, it is important to note that using artificial intelligence methods to characterise the melt-through state in laser welding process monitoring is a data-driven research tool.

In the process monitoring of aluminium alloys laser welding, we are more concerned with the macroscopic post-weld condition of the welded material, as the full penetration and non penetration state of the welding directly affects the quality of the welded aluminium alloy sheet, while the use of acoustic emission monitoring allows for non-contact, non-destructive determination of the weld condition through keyhole acoustic information.

We therefore focus more on the influence of the laser process parameters on the acoustic information generated by the keyhole of the aluminium alloy than on the chemical composition and mechanical properties of the material. Of course, we strongly agree with your suggestion and therefore we also present a new Table 2. for the composition and mechanical properties of 6061 aluminium alloys.

Point 6:

- Table 1 - why these parameters were used? It is not clear. Potential readers should have clear information from the text.

Response 6:

Thank you for your question, as we mentioned in response 3 "We therefore focus more on the influence of the laser process parameters on the acoustic information generated by the keyhole of the aluminium alloy than on the chemical composition and mechanical properties of the material", also, using the parameters in Table 1, which were determined after extensive laser experiments and process comparisons, the three process parameters correspond to non penetration, critical penetration and full penetration, which we have shown in the super depth of field photograph in Figure 2.

Point 7:

Results and discussions:

- This section requires modifications. Firsly, there is no proof that you have welded any samples. Please show some photos. Moreover, photos will proove that used welding parameters were good, and allow to create full joint.

Response 7:

Thank you for your suggestion, however we would like to clarify that the three states have been fully illustrated in Figure 2 with ultra depth of field microscope and acoustic information, please excuse the fact that we consider the ultra depth of field figure to be a better representation than the regular welded joint picture of the difference in the welding penetration state.

Point 8:

- I cannot find any scientific discussion here. Paper looks more like technical raport than scientific paper. You should compare your results with other scientific papers. It allows to underline the biggest advantages from your work. Moreover, it will proov the necessity and novelty of your work.

Response 8:

      Thank you for your question. In general, our work uses an airborne acoustic emission sensor to capture the acoustic waves emitted during laser welding of an aluminium plate and uses mathematical methods to model and predict the weld penetration. Due to the Gaussian distribution of the laser energy, the laser beam irradiates an unstable keyhole in the aluminium plate and emits metal vapour and plasma, the diffusion of the metal vapour and plasma creates a pressure wave which forms an audible acoustic wave in the form of vibration, therefore the captured acoustic wave can reflect the oscillation and variation of the keyhole.

    This paper uses time-frequency analysis methods to analyse the acoustic signal numerically and deep learning methods to train the acoustic signal as a model, which ultimately looks more like a numerical analysis, but this is necessary and relevant to the ultimate goal of penetration identification.

Certainly, any scientific study requires a great deal of comparison and analysis, and in our work we have also made a great deal of comparisons, as detailed in the parameters in new Table.3 and the accuracy comparison in Figure 15, which ultimately shows that the model proposed in this paper is more accurate and has a shorter iteration period.
The machine learning approach is now the more mainstream means of penetration identification, which is more accurate and robust, and the object of comparison using machine learning methods is usually another machine learning method rather than a traditional manual prediction method, so the object of comparison in this paper is also selected from other deep learning models, which is more comparable.

Point 9:

Conclusions:

- Please support with the quantitative results.

Response 9:

Thanks for the question, we have highlighted the quantitative results.

All the corrections you can see are in the attachment as the new manuscript 

Reviewer 3 Report

[1].  Abstract is not appropriate. Here the results which you have obtained must be highlighted, not conclusively as this is the part of conclusion.

[2].  All reference must be written in square brackets.

[3].  What are the other methods of conventional measurement systems are available for penetration monitoring?

[4].  A separate paragraph must be included in the introduction section that highlights different mechanisms and systems for the monitoring of penetration whether in process, or after welding.

[5].  I don’t encourage heading 2 and 3 separately, possibly these can be the subheading of Section 1, introduction.

[6].  Methodical approach is again heading 2, kindly correct.

[7].  How CNN and LSTM models are different from ANN? Are they requiring training of data as requires for ANN?

[8].  Why not authors compare their results with any other experimental works and go for the ML prediction, and kindly explain how much it useful and reliable is for the research community?

Author Response

Point 1:

[1]Abstract is not appropriate. Here the results which you have obtained must be highlighted, not conclusively as this is the part of conclusion.

Response 1:

Thank you for your question, we agree with you that "the abstract should express the results in the conclusion", so we have spent a lot of time in training and optimizing the deep learning model for this paper to achieve a prediction accuracy of 99.8%.

This is what we wanted to achieve in order to identify the melt-through state of laser welding with high accuracy. We have therefore respected your suggestion and have highlighted this result in the abstract revision.

Point 2:

[2]All reference must be written in square brackets.

[3]What are the other methods of conventional measurement systems are available for penetration monitoring?

[4]A separate paragraph must be included in the introduction section that highlights different mechanisms and systems for the monitoring of penetration whether in process, or after welding.

 [5]I don’t encourage heading 2 and 3 separately, possibly these can be the subheading of Section 1, introduction.

[6] Methodical approach is again heading 2, kindly correct.

Response 2:

Thank you for your suggestion. To clarify, the problems you saw with the reference formatting of this paper were due to errors caused by machine formatting, so we have reformatted it.

The corrected introductory section includes two subheadings, an introduction to acoustic emission monitoring and an introduction to machine learning methods.

Other methods for monitoring the laser welding process include visual monitoring, optical monitoring and thermal sensing monitoring, all of which are available in the literature from previous studies.

Please understand that since this paper focuses on the monitoring of acoustic emission sensing, the other methods will not be introduced to any great extent.

Point 3

[7].  How CNN and LSTM models are different from ANN? Are they requiring training of data as requires for ANN?

Response 3

Thank you for your questions and interest in our articles, we are pleased to respond to your questions. AI (AI: Artificial Intelligence) is a broad term that covers the ability of a machine to classify, predict and make decisions without human interaction. You might use some forms of AI to develop algorithms with simple if-then rules and decision trees.

As computing power increases, these algorithms are also used to control the behavior of the AI. This advance in AI has brought about subsets of ML (ML: Machine Learning) and DL (DL: Deep Learning).

The main algorithms used to drive ML and DL are called ANN (ANN: Artificial Neural Networks), which are inspired by biological neural networks in the human brain.

The most common ANN used for DL is a CNN (CNN: Convolutional Neural Network), which consists of an input and output layer and multiple hidden layers. The hidden layers operate on the inputs and then pass the results to the next layer. Each layer builds on the previous layers to improve the ability to classify information.

Another common ANN in DL is the LSTM (LSTM: Long Short Term Memory Networks), which consists of several units, including memory units, forgetting units and output units, which are formed into a single LSTM unit, and the LSTM shows strong advantages in dealing with temporal problems.

This paper combines a one-dimensional CNN with an LSTM for processing one-dimensional time-series acoustic signals to characterize the laser welding penetration state, giving full play to the CNN's ability to process spatial features, i.e. mining the frequency domain information of acoustic signals, and the LSTM's ability to process time-series problems, i.e. mining the time-domain information of acoustic signals, resulting in the construction of a spatial-temporal neural network framework.

Such a framework is very relevant and necessary for the processing of acoustic emission sensing signals that contain complex acoustic information.

Point 4

[8].  Why not authors compare their results with any other experimental works and go for the ML prediction, and kindly explain how much it useful and reliable is for the research community?

Response 4

Thank you for your question, we understand your concern, any scientific research requires a lot of comparisons and analysis.

In our work, we have also carried out a number of comparisons, which are shown in Table.2 for the comparison of parameters and Figure 15 for accuracy, and the final results show that the model proposed in this paper is more accurate and has a shorter iteration period.

The machine learning approach is now the more mainstream method for penetration identification, which is more accurate and robust, and the object of comparison using machine learning methods is usually another machine learning method rather than a traditional manual prediction method, so the object of comparison in this paper is also selected from other deep learning models, which is more comparable and convincing.

Please see all the corrections in the attachment as our covered manuscript.

Round 2

Reviewer 2 Report

Dear Authors,

Many of my comments and questions are not solved. I suggest major revision once again.

1. Please add the QUANTITATIVE results into the abstract.

2. "- You have not presented any issues about welding of used base material. Yopu should show, which methods could be used, which problems could be observed. It should underline, that your method could be used as the best."

Still, no information is presented in INTRODUCTION. Which method are used for joining your BM? Which problems could be observed? These issues should show, that laser welding provides to biggest advantages in joining. It will underline the necessity oy your investigations.

3. Mechanical properties (yield point, tensile strength, elongation) of used materials are not presented. Potential readers shoud get clear, full information about BM from your paper.

4. Table 2 - the source of presented values is unknown. Have you tested chemical composition? If yes,, which method(s) was/were used? If values are taken from standard/manufacturer data/other source, the source should be presented in the paper. Moreover, this table is not mentioned in the text.

5. Please show the top view of your specimens. It will proove that no surface imperfections occured. Not only penetration proved propper chosen of parameters.

6. "You should compare your results with other scientific papers. It allows to underline the biggest advantages from your work. Moreover, it will proov the necessity and novelty of your work."

This comment still exists. Please include this type of scientific discussion.
